# Order–order assembly transition-driven polyamines detection based on iron−sulfur complexes

Yahui Zhang [1], Xiangyu Zhao[2], Yue Qin[1], Xiaopei Li[1], Yongxin Chang[1], Zhenqiang Shi[1], Mengyuan Song[1], Wenjing Sun[1], Jie Xiao[1], Zan Li[1] & Guangyan Qing [1,3 ✉]

Innovative modes of response can greatly push forward chemical sensing processes and subsequently improve sensing performance. Classical chemical sensing modes seldom involve the transition of a delicate molecular assembly during the response. Here, we display a sensing mode for polyamine detection based on an order–order transition of iron–sulfur complexes upon their assembly. Strong validation proves that the unique order–order transition of the assemblies is the driving force of the response, in which the polyamine captures the metal ion of the iron–sulfur complex, leading it to decompose into a metal–polyamine product, accompanied by an order–order transition of the assemblies. This mechanism makes the detection process more intuitive and selective, and remarkably improves the detection efficiency, achieving excellent polyamines specificity, second-level response, convenient visual detection, and good recyclability of the sensing system. Furthermore, this paper also provides opportunities for the further application of the iron–sulfur platform in environment-related fields.

[1] Key Laboratory of Separation Science for Analytical Chemistry, Dalian Institute of Chemical Physics, Chinese Academy of Sciences, 457 Zhongshan Road, Dalian 116023, China. [2] Sixth Laboratory, Sinopec Dalian (Fushun) Research Institute of Petroleum and Petrochemicals, 96 Nankai Road, Dalian 116045, P. R. China. [3] College of Chemistry and Chemical Engineering, Wuhan Textile University, 1 Sunshine Road, Wuhan 430200, P. R. China. ✉email: qinggy@dicp.ac.cn

The detection of organic amines has received special attention for their key roles ranging from organic synthesis[1] to environmental monitoring[2]. Usually, organic amines are highly toxic, harmful, colorless, and without chromogenic groups, which renders them difficult to be directly differentiated through visual inspection[3–5]. Thereinto, polyamines are implicated in diverse functional applications, such as analytical detection[6], drug development[7], chemotherapy[8–10], and biological regulation[11], which makes the determination of polyamines particularly important[12,13], especially in synthetic polyamines detection. For example, the melamine scandal, as an infant-formula milk scandal that results in the hospitalization of tens of thousands of children and even a few deaths[14], has alarmed the safety of global food chains[15]. By far, various sensing materials have been involved in the polyamines detection fields, including molecular probes[16,17], chemosensors[18,19], and nanosensors[20,21], usually at a certain temperature, reaction time, and pH conditions[22]. From the perspective of sensing mechanism, the above could be categorized into two types of response modes. One is the direct bonding between the sensor and the analyte, including irreversible covalent-like and reversible noncovalent bondings. The other is the indicator displacement between the indicator (ligand) and the target analyte, both of which mainly focus on the molecular level interaction. By comparison, seldom works report the chemical sensing based on molecule self-assembly[23]. Due to the more remarkable changes at supramolecular level, the formation or destruction of the ordered assemblies during the response process could greatly facilate the sensing efficiency, however, it is difficult to be manipulated precisely. Here, beyond these transitions from order to disorder, we innovatively propose a selective polyamines detection mode driven by an order–order transition of the assemblies based on an iron–sulfur platform.

Iron–sulfur complexes, inspired by biological enzyme cofactors, exhibit powerful tasks in terms of their unique activity and selectivity, such as catalysis[24] and electron-transfer[25]. Over decades, a variety of artificial iron–sulfur complexes[26–31] with core skeletons composed of iron atoms, sulfur–donor ligands, and auxiliary ligands are developed. However, these synthetic highly active model complexes are usually characterized by low coordination numbers[32] and low metal valence states[33]. Their inherent fragility and reactivity to air and water dictate that all related operations should be performed under strict anhydrous and anaerobic conditions[24,34], which greatly limits their application in the environment. Nevertheless, the distinctive redox and coordination properties of the iron–sulfur moiety also offer the perspective of developing unknown applications in the sensing detection, and the key is to design and synthesize a structurally stable iron–sulfur system.

In this paper, an iron–sulfur platform act as universal polyamines sensors was first developed and a unique order–order assembly transition mechanism based on a self-sacrificial reaction was revealed smoothly. Initially, the environmentally stable mononuclear complex [Cp*Fe($\eta^3$-tpdt)] (tpdt = S[CH$_2$CH$_2$S$^-$]$_2$, 3-thiapentane-1,5-dithiolate) composed of a sterically large pentamethylcyclopentadienyl (Cp* = $\eta^5$-C$_5$Me$_5$) ligand[35–37] and a flexible tpdt ligand. On this basis, diverse thiolate-bridged homo-/ heteronuclear complexes were then constructed with the interactions of transition metal ions Zn$^{2+}$, Fe$^{2+}$, Mn$^{2+}$, Pd$^{2+}$, Ni$^{2+}$, and Cu$^+$, induced by M−S (M = metal) d-p π conjunction (Fig. 1). Taking iron-zinc complex as model sensor, this sensing material could specific recognize different kinds of polyamines (especially for diethylenetriamine and cyclen) through ultraviolet-visible (UV-vis) spectroscopy or naked-eye detection, and a low polyamines detection limit of 1 μM was achieved *via* differential pulse voltammetry (DPV) method. Thus, we found that an order–order assembly transition during the response is the

driving force, in which highly ordered self-assembled morphologies were discovered in this sensing material at both microscopic and single-crystal molecular levels. Meanwhile, an order–order transition of the assemblies occurred after the interaction of the binuclear Fe–Zn complex with a polyamine, accompanied by the release of the mononuclear iron precursor and the generation of Zn-polyamine product (Fig. 1). Importantly, this sensing material could also be rapidly recovered and then reused again through the addition of zinc chloride, and the number of cycles reaches more than ten times. Surprisingly, we found that this array composed of different transition metals possessed different responses to polyamines, monoamines, as well as other interfering compounds, and further achieved specific recognition between them. Moreover, the driving force detection and the recycling availability were in common with these sensors. Such an iron–sulfur platform not only proposes a unique polyamines detection mechanism, but also discloses the huge development potential of iron–sulfur complexes in sensing applications.

## Results

**Synthesis of iron–sulfur complexes**. As depicted in Fig. 2, environmentally stable mononuclear complex [Cp*Fe($\eta^3$-tpdt)][38] was reacted with 1 equiv. of ZnCl$_2$ in CH$_2$Cl$_2$ for 1 h at room temperature, which gave a purplish-red solution of heterobinuclear complex [Cp*Fe($\mu$-1$_\kappa^3$SSS':2$_\kappa^2$SS-tpdt)ZnCl$_2$] (**C1**). Furthermore, binuclear complexes [Cp*Fe($\mu$-1$_\kappa^3$SSS':2$_\kappa^2$SS-tpdt) MCl$_2$] (**C2**, M = Fe; **C3**, M = Mn) and heterotrinuclear complexes [Cp*Fe($\mu$-1$_\kappa^3$SSS':2$_\kappa^2$SS-tpdt)M($\mu$-2$_\kappa^2$SS:3$_\kappa^3$SSS'-pdt) FeCp*][PF$_6$]$_n$ (**C4**, M = Pd, $n = 2$; **C5**, M = Ni, $n = 2$; **C6**, M = Cu, $n = 1$) were also successfully synthesized through the interaction of [Cp*Fe($\eta^3$-tpdt)] with different metal salts, regardless of the concentrations of the iron precursor and metal salts. Following purification, complexes **C1**−**C6** were all obtained in good isolated yields (90%–95%). They all displayed excellent chemical stability in air and moisture, judged by proton nuclear magnetic resonance ($^1$H NMR) data.

Besides, the introduction of other metal ions into the mononuclear iron system was also explored. Ca$^{2+}$, Mg$^{2+}$, and lanthanide elements (e.g., La$^{3+}$, Sm$^{3+}$, Eu$^{3+}$, Ce$^{3+}$) did not coordinate with sulfur atom even after heating under reflux. By contrast, obvious reaction phenomena (color changes) were observed between iron precursor and Al$^{3+}$, Ti$^{4+}$ or V$^{3+}$, but their final products exhibited too poor stability in the air, which largely limits their applications in the environment.

**Characterization of C1−C6**. Molecular structures of complexes **C1**−**C6** were confirmed by single-crystal X-ray diffraction analysis (**C1**: CCDC 2171251, see Supplementary Data 1 and 6 for details; **C3**: CCDC 2171252, see Supplementary Data 2 and 7 for details) and their corresponding ORTEP drawings are shown in Supplementary Figs. S1–9. The details of the data collection and refinement are presented in Supplementary Tables S1 and S2. The main bond lengths and bond angles are shown in Supplementary Tables S3–8. Moreover, magnetic properties and outer electron distribution status of complexes **C1**−**C6** were characterized by $^1$H NMR (Supplementary Fig. S16), electron paramagnetic resonance (EPR) spectroscopy (see Supplementary Methods, Figs. S21–25 for details), and Fourier transform infrared (FT-IR) spectroscopy (Supplementary Figs. S49, S50). All complexes were paramagnetic except for **C5**. EPR spectrum of complexes **C1** (Supplementary Fig. S21) and **C3**[39,40] (Supplementary Fig. S23) in CH$_2$Cl$_2$ exhibited strong signals at g = 2.10 and 2.08, respectively, further affirming the involvement of the free radicals. Electrospray Ionization High Resolution Mass Spectrum (ESI-HRMS) data of

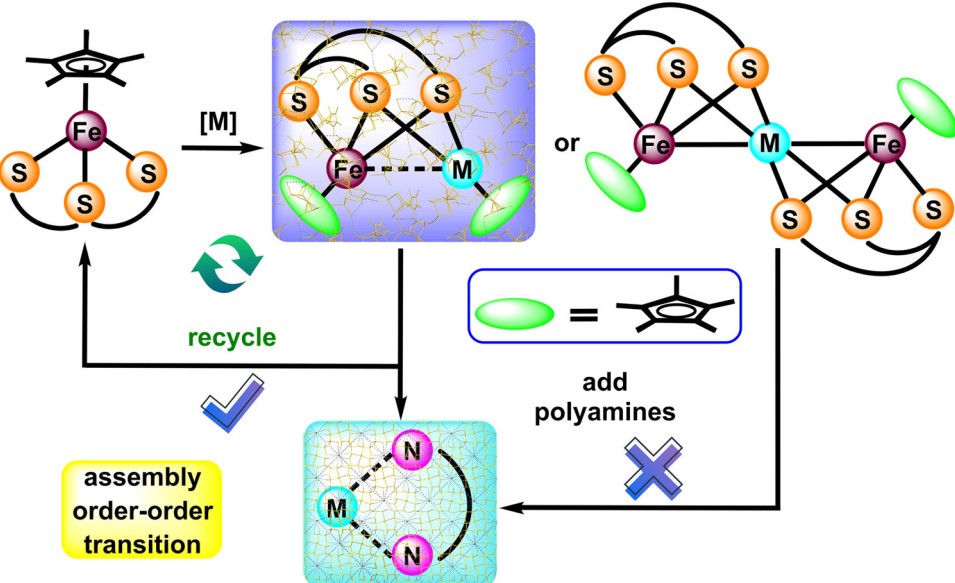

**Fig. 1 Reaction design for selective polyamines detection.** Graphic illustration of the selective, reversible polyamines detection strategy based on an iron–sulfur platform, driven by an order–order transition of the assemblies from a binuclear iron-metal sensor to a metal-polyamine product upon the addition of polyamine.

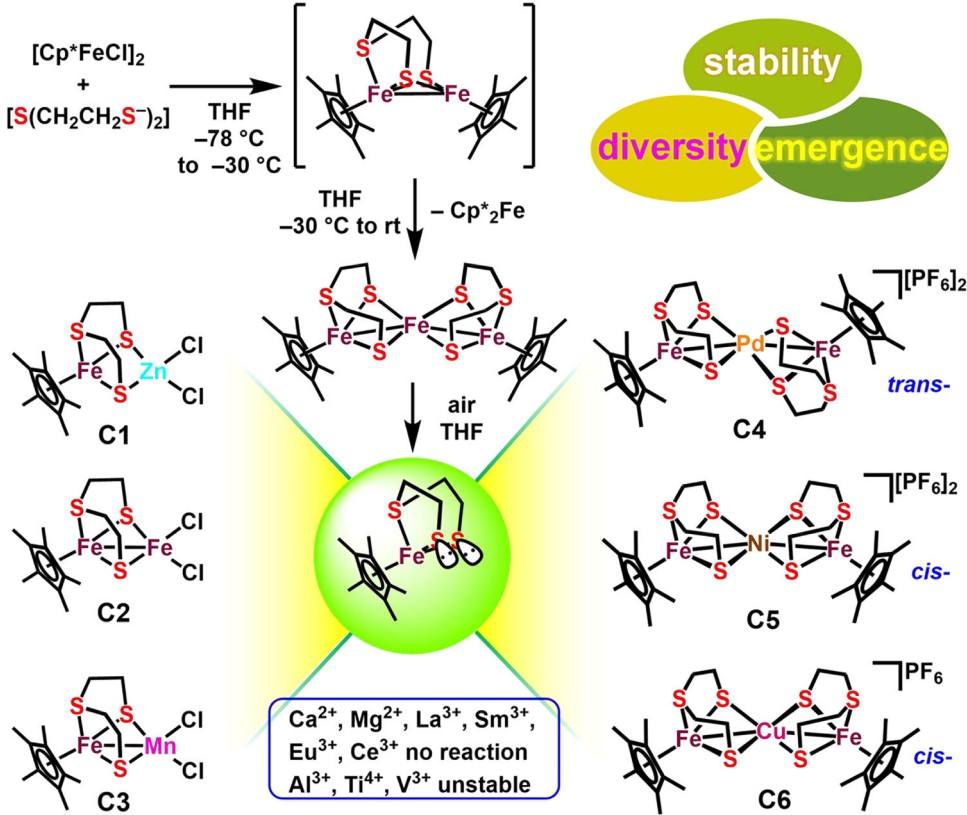

**Fig. 2 Design strategy of polyamine sensors containing iron–sulfur moiety.** Synthetic route of complexes **C1**–**C6** using monoclear [Cp*Fe($\eta^3$-tpdt)] as the precursor.

**C4**–**C6** were also smoothly characterized obtained in the solution state (Supplementary Figs. S42–44).

In addition, complexes **C1**–**C6** presented a wide range of colors in $CH_2Cl_2$, from yellow to purplish-red, and the photos captured in daylight are depicted in Supplementary Fig. S27A. The UV-vis spectra (see Supplementary Methods for details) of **C1**–**C6** with

the precursor [Cp*Fe($\eta^3$-tpdt)] had been shown in Supplementary Fig. S27B, C. The UV-vis spectrum of the precursor [Cp*Fe($\eta^3$-tpdt)] exhibited two intense absorption bands around 420 and 588 nm. These two bands were maintained for binuclear complexes **C1**–**C3** and experienced different degrees of a blue shift in **C1** (385, 554 nm) and **C3** (395, 521 nm), while **C2**

(640 nm) featured a redshift in the visual region (Supplementary Fig. S27B), which manifested a strong molecular metal-to-metal charge transfer character[41]. In the UV-vis spectra of C4–C6, the electron delocalization[42,43] between the three metal atoms led to the disappearance of the original two bands and the formation of a stronger peak near 500 nm (Supplementary Fig. S27C).

**Responsive behaviors to polyamines**. Then responsiveness capacity of C1–C6 to polyamines was investigated by using the UV-vis absorption spectra before and after the addition of various reagents. Initially, the UV-vis spectra of precursor [Cp*Fe($\eta^3$-tpdt)] were explored and showed no credible changes in response to various organic amines (e.g., diethylenetriamine, cyclen, 4,4'-bipyridine, and $^n$propylamine), even in the presence of high amines concentrations up to 15 mM (Fig. 3A). Interestingly, C1 displayed a remarkable response to diethylenetriamine and cyclen, and the differences in magnitude indicated the selective response of C1 to different polyamines (Fig. 3B, C). With increasing concentrations of diethylenetriamine or cyclen from 0.05 to 0.30 mM, the absorption band at 590 nm of C1 gradually increased and the peak at 385 nm showed a 34 nm redshift. In contrast, C1 was silent for low concentrations of 4,4'-bipyridine (Fig. 3D) or $^n$propylamine (a typical monoamine, Fig. 3E), and only the high concentrations (>2 mM) of analyte could result in considerable changes in their UV-vis spectra. No change was observed in the presence of ethyl acetate (EtOAc) at the concentration of 15 mM (Fig. 3F). The above results revealed that the introduction of Zn$^{2+}$ into the mononuclear iron–sulfur skeleton endowed satisfactory sensitivity, selectivity, and specificity of C1 toward polyamines.

Analogous polyamine selectivity was also detected in C2, the addition of diethylenetriamine or cyclen (0.40 mM) led to an obvious blue-shift of the absorption peak at 642 nm, which was different from the occurrence of only small or no variation upon the addition of 4,4'-bipyridine, $^n$propylamine or EtOAc (0.40 mM) (Fig. 3G). In contrast, iron-manganese complex C3 could respond to both polyamines and monoamines, and no credible variation was detected upon the addition of EtOAc (Fig. 3H), indicating a high selectivity of C3 toward amine species. The trinuclear complexes C4, C5, and C6 exhibited no photophysical response toward various amines and EtOAc (Supplementary Figs. S34B, C, G, S35D, S36B, C, G, S37D, S38B, C, G, and S39D).

In addition to the remarkable variation in the UV-vis spectra, an interesting phenomenon attracted our attention. Significant elevation of the baseline of C1, C2, or C3 was observed upon the addition of diethylenetriamine or cyclen, as shown by the blue dash lines in Fig. 3B, C, 3G, H. Meanwhile, several tiny particles were formed, accompanied by the color change from purplish-red to green in the solution samples. This interesting phenomenon were confirmed by the "Tyndall" effect[44,45] (Fig. 3B, C insets), a light scattering phenomenon observed in colloidal suspensions, which lies at the core of this work and will be elaborated on later.

**Visible colorimetric assay**. Benefiting from the selectivity and specificity of the above complexes responding toward various amines as well as the obvious color change of the sample solution, a colorimetric visible assay[46,47] could be easily built. Figure 4A shows the colorful host solutions of the iron–sulfur precursor and complexes C1–C6. No color change was observed in the presence of EtOAc (Fig. 4B). When $^n$propylamine (Fig. 4C) or 4,4'-bipyridine (Fig. 4D) was added to each complex solution, only C3 immediately changed from purple to green, which revealed the superiority of C3 in rapid amine species detection. When cyclen (Fig. 4E) or diethylenetriamine (Fig. 4F) was used as the analyte,

the colors of complexes C1, C2, and C3 solution all immediately changed to green, despite their different initial colors. In these tests, the iron–sulfur precursor and complexes C4–C6 showed no color changes in response to these analytes. The above results indicated that the colorimetric assay allowed for the quickly naked-eye discrimination of polyamines from monoamines and non-amines interferences, and could specifically distinguish different polyamines.

**Amperometric detection of polyamines**. Specific recognition to different kinds of polyamines have been achieved through portable visible colorimetric assay, but more sensitive and specific polyamine detection could be achieved through the DPV tests[48,49]. As shown in Fig. 4G, with increasing cyclen concentrations of 60 to 1800 μM, DPV intensity gradually decreased (see Supplementary Methods for details). According to the reduction of the current size at −0.96 V, a calibration plot between the absolute values of the current intensity of C1 and cyclen concentrations was established (red line in Fig. 4H). The fitting linear regression formula was defined as $I$ (μA) $= -0.00921\ C_{cyclen}$ (μM) $+ 17.6$ ($R^2 = 0.989$) and a limit of detection (LOD) was calculated as $4.3 \times 10^{-6}$ M using the equation 3 $N/S$ ($N$, the standard deviation of the blank samples; $S$, the slope of the fitting line). Surprisingly, the micromolar detection concentration of the LOD calculated by the above equation was consistent with the minimal allowed concentration in the actual measurement. Compared to the silence in the presence of 0.3 μM cyclen, the addition of 3 μM cyclen could cause a small change in the DPV spectrum of C1 (Supplementary Fig. S47A). Meanwhile, the different slopes of the black line (diethylenetriamine, −0.01198), the red line (cyclen, −0.00921), and the green line (4,4'-bipyridine, −0.00366) indicated that C1 could accurately distinguish between different types of polyamines (Fig. 4H, Supplementary Fig. S47B, C). Under the same condition, C1 exhibited no significant response to $^n$propylamine (blue triangle in Fig. 4H and Supplementary Fig. S47D). All in all, the specific recognition of different polyamines with a lower detection limit and higher sensitivity was successfully achieved by the comparison of the slopes, confirming that C1 was an excellent sensor.

**Response mechanism research**. Here, complex C1 and cyclen were selected as the model material and analyte to elaborate on the sensing mechanism, and a series of spectroscopic tests were performed. Upon the addition of cyclen, the characteristic EPR signal at the spectral splitting factor $g = 2.10$ shifted to 2.09 (Fig. 5A), indicating the formation of a original compound with a free radical. In the Cyclic Voltammogram (CV) test (Fig. 5B), complex C1 (black line) displayed a quasi-reversible peak at $E_{1/2} = -0.957$ V and an irreversible oxidation peak at $E_p^{ox} = 0.816$ V versus Ag/AgCl, while no redox peak was observed in the CV spectrum of pure cyclen (red line). After the mixing of the two at a molar ratio of 1:1, the characteristic quasi-reversible signal of C1 disappeared and an oxidation peak at 0.073 V appeared (blue line), which was consistent with the characteristic peak of the iron–sulfur precursor [Cp*Fe($\eta^3$-tpdt)] (Supplementary Fig. S47B). Furthermore, in the $^1$H NMR titration experiment (Fig. 5C), individual C1 showed two broad peaks at −9.91 and −32.24 ppm, attributing to the protons of S-CH$_2$ and Cp*-CH$_3$. With the addition of cyclen, these two peaks shifted to −6.63 and −26.03 ppm, respectively, which was consistent with the data of the precursor [Cp*Fe($\eta^3$-tpdt)] (Supplementary Fig. S47C). Considering the data of EPR, CV, and $^1$H NMR, we presumed that cyclen captured Zn$^{2+}$ in C1, which led to the dissociation of the binuclear complex C1 into mononuclear [Cp*Fe($\eta^3$-tpdt)] (Supplementary Fig. S48). Therefore, a self-

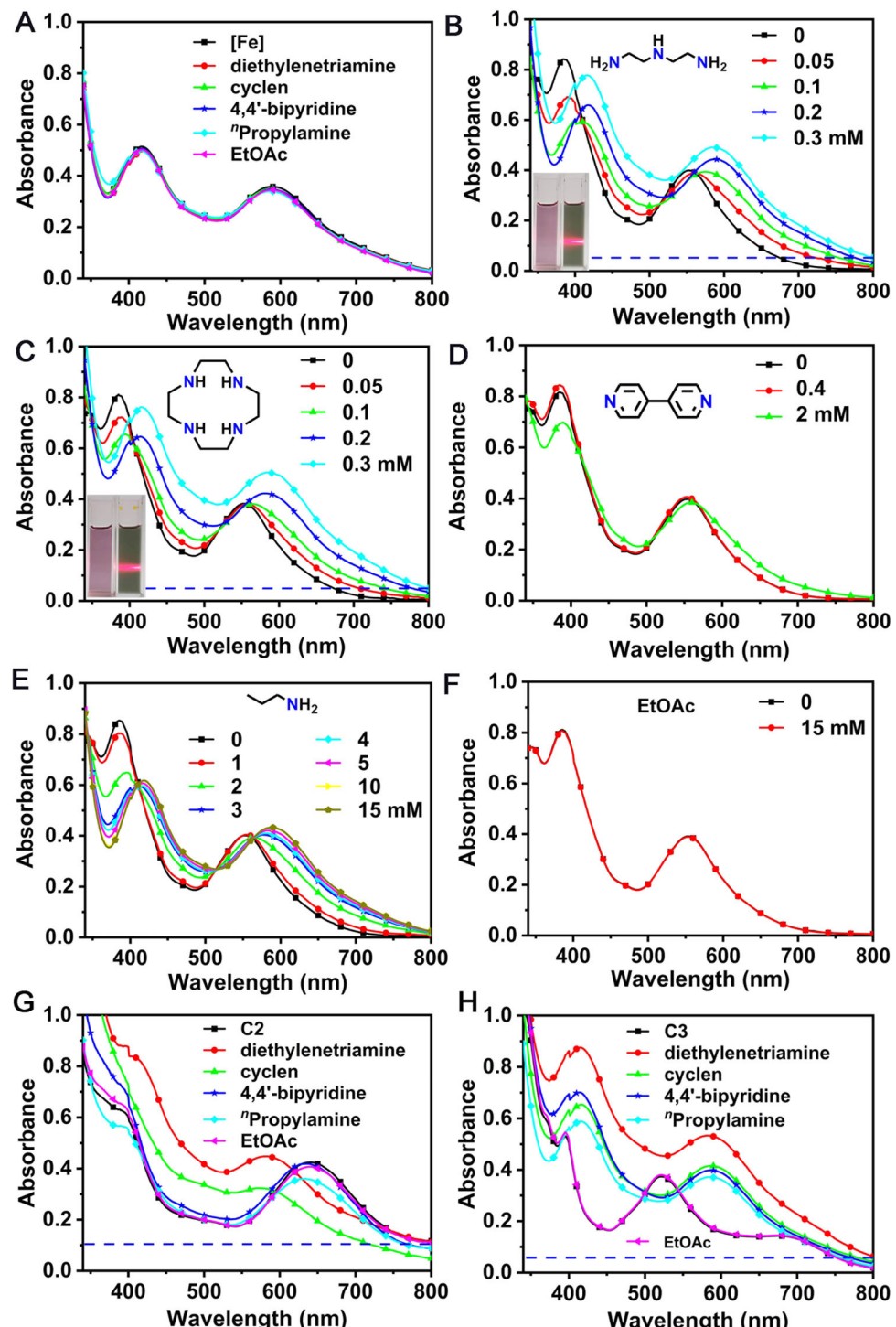

**Fig. 3 UV-vis absorption spectra tests. A** UV-vis spectra of precursor [Cp*Fe($\eta^3$-tpdt)] (abbreviated as [Fe], 0.33 mM) upon additions of various reagents (15 mM). **B–F** UV-vis spectra of **C1** recorded during (**B**) diethylenetriamine (0 to 0.3 mM), (**C**) cyclen (0 to 0.3 mM), (**D**) 4,4'-bipyridine (0 to 2 mM), (**E**) $^n$proylamine (0 to 15 mM), or (**F**) EtOAc (0 to 15 mM) were added to the CH$_2$Cl$_2$ solution of complex **C1** (0.33 mM). Tyndall effect of **C1** (0.33 mM) in the presence of diethylenetriamine or cyclen at the concentration of 0.30 mM was inserted in **B** and **C**, respectively. **G, H** UV-vis spectra of complexes **C2** (**G**) and **C3** (**H**) upon additions of various reagents (0.40 mM).

sacrificial reaction rather than conventional binding is a possible mechanism underlying the cyclen recognition process (Fig. 1).

During the polyamines detection process, changes in system composition could affect the magnetic resonance relaxation[50] of the whole system. Then low-field $^1$H NMR spectroscopy was performed to measure the spin-spin relaxation time ($T_2$) of protons, and a smaller $T_2$ value corresponded to a stronger

influence of the paramagnetic compound on the whole system. The decay curves imply that the $T_2$ (measured by a Carr–Purcell–Meiboom–Gill (CPMG) sequence)[51,52] of **C1** and cyclen in CH$_2$Cl$_2$ are 987 and 3388 ms, respectively, while the $T_2$ of their mixture decreases to 689 ms (Fig. 5D). These data indicated that the effect of cyclen on $T_2$ was almost negligible for the whole system and the change in the complex after the self-

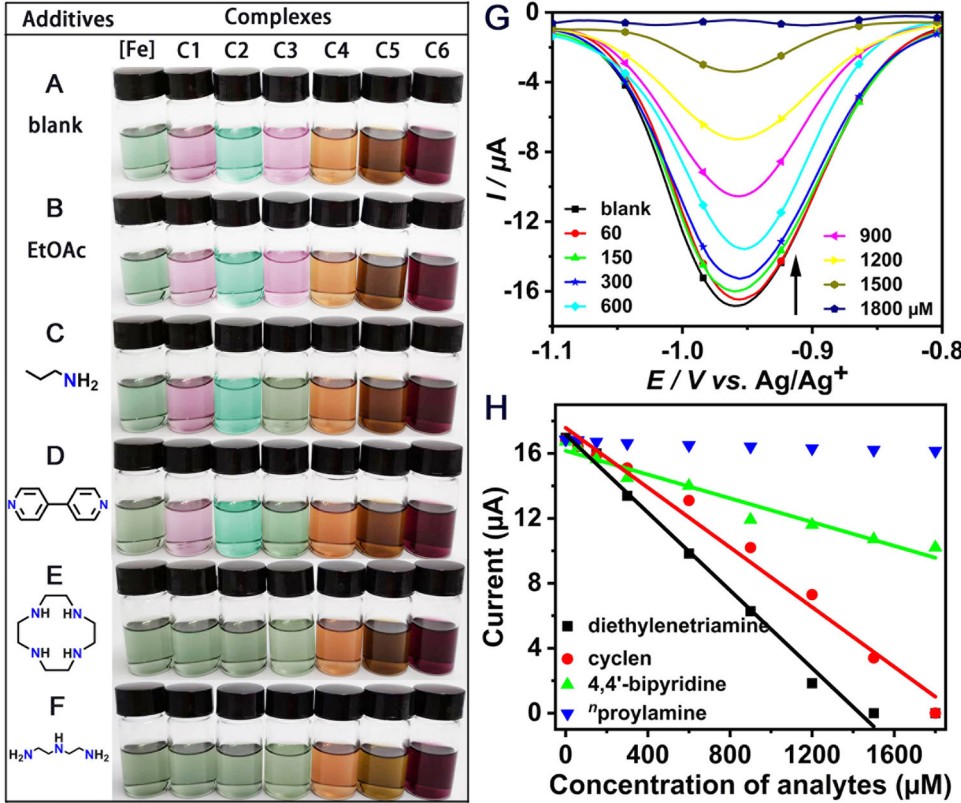

**Fig. 4 Selective and specific detection of iron–sulfur array to polyamines.** A–F Comparison of all iron-based complexes (0.33 mM) in terms of color changes after the additions of various reagents (0.40 mM). G DPV responses of **C1** (1 mM) toward various concentrations of cyclen (60, 150, 300, 600, 900, 1200, 1500, and 1800 μM) in the CH$_2$Cl$_2$ solution with 0.1 M $^n$Bu$_4$NPF$_6$ as electrolyte. Potentials were referenced versus Ag/AgCl. H Calibration plots of **C1** for diethylenetriamine (black), cyclen (red), 4,4'-bipyridine (green), and $^n$propylamine (blue) detection based on the reduction peak at −0.96 V of DPV data, respectively.

sacrificial reaction was the main determining factor. The generated mononuclear [Cp*Fe($\eta^3$-tpdt)] exhibited a stronger magnetism than **C1**. According to the "Tyndall" effect depicted in Fig. 3B, C insets, a dynamic light scattering (DLS) test was implemented, the details could be found in Supplementary Methods. The top histogram in Fig. 5E indicates that there are numerous tiny nanoparticles (NPs) with an average diameter of 11 nm in the complex **C1** solution, which sharply increased to 602 nm after cyclen addition (the bottom histogram).

**Order–order assembly transition.** To determine the NPs composition, one droplet of **C1** solution in CH$_2$Cl$_2$ was dropped onto the silicon wafer to test for the morphology and elemental analysis by scanning electron microscopy (SEM, see Supplementary Methods for details). As shown in Fig. 6A, numerous irregular block-shaped particles with an average size of 2.1 μm (Supplementary Fig. S54A) were observed. The elemental analysis *via* energy-dispersive spectroscopy (EDS) mapping revealed that these NPs contained C, Fe, S, Zn, and Cl, consistent with the **C1** structure. It is worth noting that the sizes of NPs detected by SEM were larger than those in the solution because the molecules were more inclined to aggregate and self-assemble on the surface during solvent evaporation[53,54]. Furthermore, the spatial packing mode of **C1** was determined by single-crystal diffraction. Each of the four **C1** molecules with different orientations formed a repeating unit and was interconnected by intermolecular H···Cl bonds with bond lengths ranging from 2.6 to 2.9 Å. None of the four Cp* in a repeating unit was coplanar and the angles between them (indicated by planes with different colors) are listed at the bottom of Fig. 6B. Furthermore, as viewed from y- and z-axes, a

**C1** molecule interlocked the other two molecules through intermolecular hydrogen bonding interaction between tpdt-H and the coordinating chlorine atom (Supplementary Fig. S2). These data indicated the highly ordered self-assembly of **C1**.

Upon the addition of cyclen to the **C1** solution, the mixture was immediately dropped onto a silicon wafer to prepare an SEM sample. As shown in Fig. 6C, the original irregular NPs disappear, instead, numerous rectangular nanosheets with an average length of 1.4 μm (Supplementary Fig. S53B) and a width of 0.9 μm (Supplementary Fig. S53C) could be detected in large areas. According to the SEM-EDS mapping, the characteristic elements of the nanosheets contained only Zn, Cl, C, and N atoms, surprisingly, while Fe and S elements were not detected. To define the exact structure of the tiny NPs, the mixture of **C1** and cyclen in CH$_2$Cl$_2$ was filtered and the residue was washed three times with $^n$hexane to generate a pale yellow solid [($\eta^4$-cyclen)ZnCl][Cl] (**N1**). Powder X-ray diffraction (XRD) spectrum of **N1** displays numerous sharp and intense reflection peaks in the $\theta$2 range between 10º and 60º. $^1$H, $^{13}$C NMR spectroscopic data (Supplementary Figs. S17 and S18), ESI-HRMS data (Supplementary Fig. S45) and FT-IR (Supplementary Fig. S51) of **N1** are also smoothly obtained. These characteristics were distinctly different from those of **C1** (Fig. 6D), which reveals the remarkable difference between **N1** and **C1** in structural composition and spatial packing mode. Subsequently, a single-crystal structure of **N1** (CCDC 2171267, see Supplementary Data 3 and 8 for details) was obtained from the CH$_3$OH/Et$_2$O solvent system and its ORTEP drawing is displayed in Fig. 6E. Four nitrogen atoms of cyclen were all coordinated to the central Zn$^{2+}$ with Zn−N bond lengths of 2.150(3), 2.152(3), 2.153(3), and 2.155(3) Å

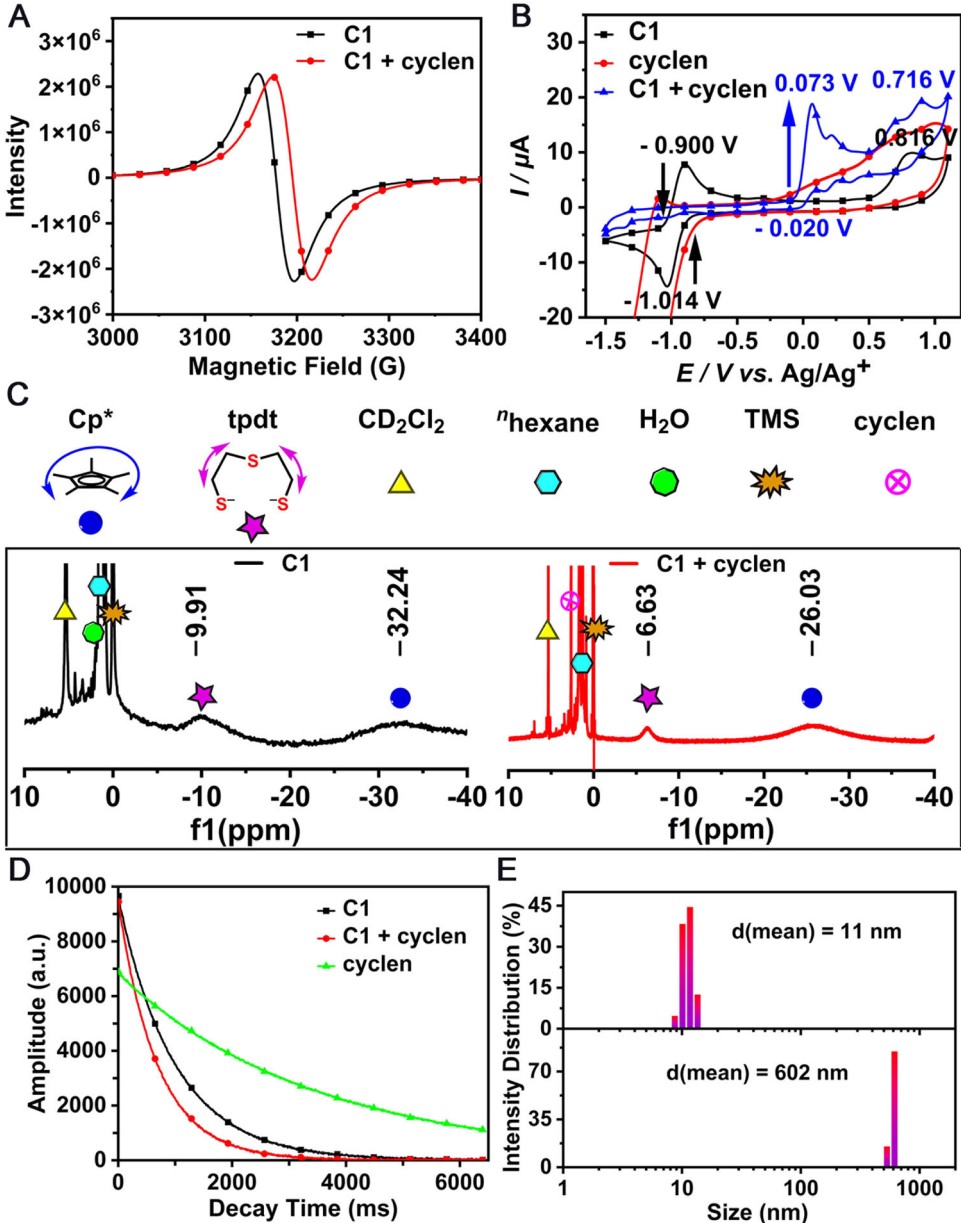

**Fig. 5 Test spectra of C1 in the absence and presence of cyclen.** EPR (**A**), CV (**B**), $^1$H NMR (**C**), low-field $^1$H NMR spectra (**D**), and hydrated diameter distribution (**E**) of **C1** (1 mM) before and after cyclen addition (1.2 mM) at room temperature. All tests were performed using $CH_2Cl_2$ except for the $^1$H NMR spectra were collected using $CD_2Cl_2$.

(Supplementary Table S9), respectively, while one Cl atom was linked to Zn forming a vertical configuration with cyclen and the other Cl atom was independent of the main structure. Although $Zn^{2+}$ complexes coordinated with cyclen[55,56] have been frequently used in the fields of catalysis, medicine, sensor, and separation, this study is the first to elucidate cyclen structure with a metal.

**N1** featured a highly ordered packing mode driven by intermolecular H···Cl bonds (Fig. 6F). Only the free chlorine atoms participated in the accumulation, to form a larger range of hydrogen bonds from 2.2 to 3.2 Å with the hydrogen atoms of the cyclen ligand. The coordinated chlorine atoms with $Zn^{2+}$ were independent of the packing model. Along the z-axis, a large grid of infinite expansion was clearly observed, composed of 12-element ring **N1** complexes and a 16-element ring centered on the free chlorine atoms (Supplementary Fig. S11A). Moreover, a regular arrangement of square cells formed by hydrogen bonds

was arranged throughout the mesh, and each square cell contained an **N1** molecule (Supplementary Fig. S11B). The single-crystal structure data of **N1** well explained the origin of nanosheets observed before, while clearly displaying what happened in the reaction solution.

In brief, the addition of cyclen captured $Zn^{2+}$ in **C1**, leading to the decomposition of binuclear complex **C1** into mononuclear [Cp*Fe($\eta^3$-tpdt)]. Under this condition, the ordered self-assembly of **C1** was destroyed, in which the intra- or intermolecular hydrogen bonds formed by the coordinated chlorine atoms with tpdt and Cp* were disrupted, accompanied by the removal of the iron precursor [Cp*Fe($\eta^3$-tpdt)] and the freeing of one chlorine atom. Subsequently, cyclen was coordinated to the $Zn^{2+}$ and formed a hydrogen bond grid with the free chlorine atom. Importantly, the newly formed $Zn^{2+}$−cyclen complex exhibited a stronger self-assembly capacity than **C1**, generating a large number of nanosheets.

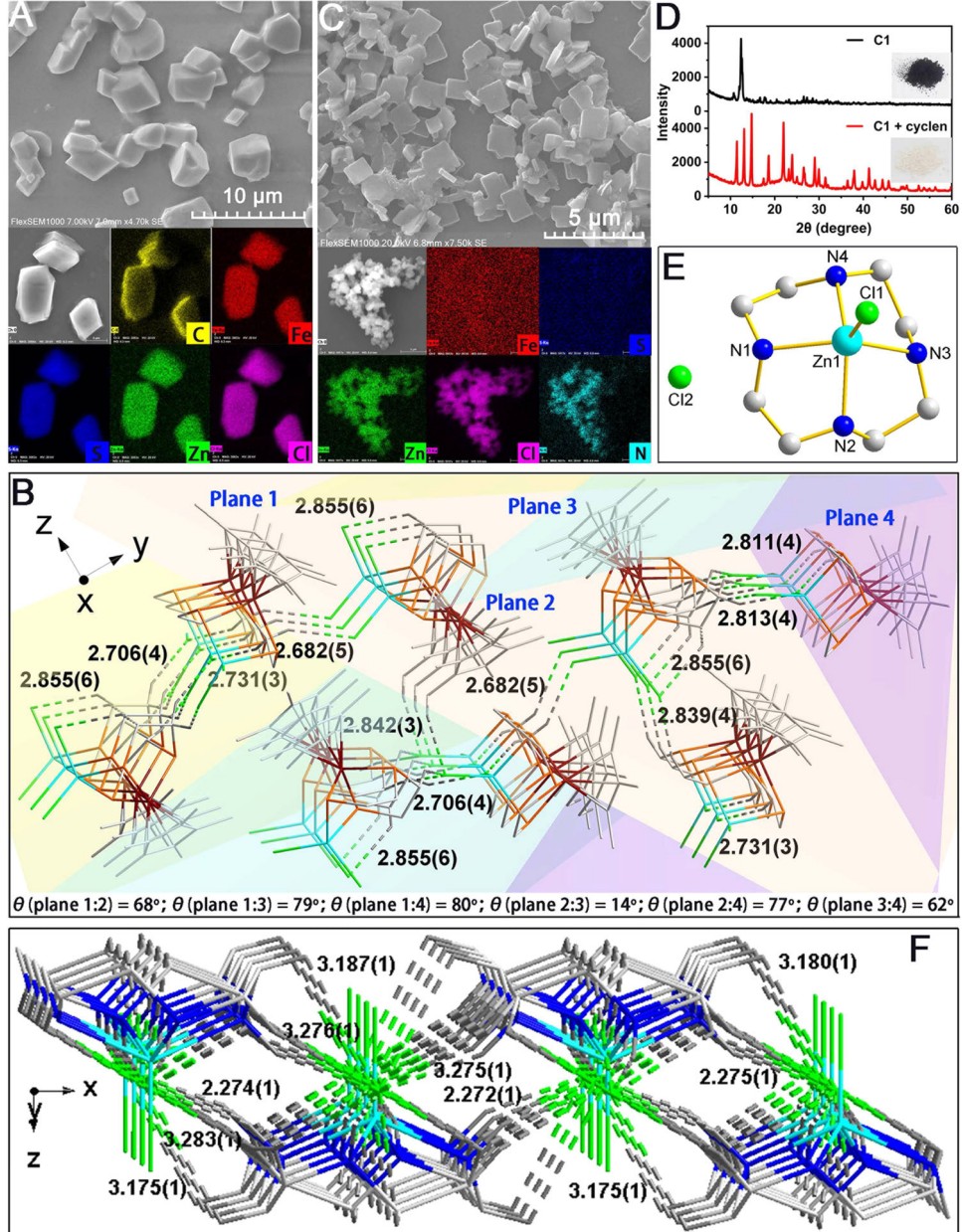

**Fig. 6 Characterization of C1 and N1.** SEM images and corresponding SEM-EDS elemental mapping images of **C1** (5.0 mM) in CH$_2$Cl$_2$ before (**A**) and after (**C**) cyclen addition (6.0 mM). **B** Crystal packing map of **C1** (CCDC 2171251) along the *x*-axis. XRD pattern (**D**) and crystal structure (**E**) of as-synthesized complex **N1** (CCDC 2171267). Thermal ellipsoids are shown at the 50% probability level and hydrogen atoms are omitted for clarity. **F** Packing mode of **N1** along the y-axis.

**Recyclability of the sensing material**. A series of mechanistic research smoothly revealed the occurrence of a self-sacrificial reaction and an order–order assembly transition (Fig. 7A). In addition to the sensitive response of complex **C1** to cyclen and the unique mechanism, **C1** was recycled to detect cyclen. A simple operation flow chart is illustrated in Fig. 7B. A purplish-red solution of **C1** in CH$_2$Cl$_2$ was immediately converted into a green mononuclear complex [Cp*Fe(η$^3$-tpdt)] in the presence of cyclen, accompanied by the formation of several tiny particles. The solid-liquid mixture was separated by centrifugation, and a layer of pale yellow solid (**N1**) was clearly deposited at the bottom of the centrifuge tube. Upon the addition of ZnCl$_2$, the green solution again immediately changed to a purplish-red (**C1**) color. Afterward, the sample was reused for cyclen detection. A more visually vivid manipulation video is provided as Supplementary Movie S1. The

detection process of **C1** with the altering additions of cyclen and ZnCl$_2$ was cycled at least ten times (Fig. 7C). The corresponding kinetic detection process of **C1** toward cyclen was also recorded by a UV-vis test at 600 nm (Fig. 7D) and the rapid detection process was clearly observed (within 1 min at 20 rpm agitation and 3 min even without agitation), which exhibited the absolute advantage compared with the 30 min test time in the previous work[57].

**The specificity of polyamines detection**. To explore the specificity of organic amines detection objects, a total of 41 compounds in three categories, including chain and ring polyamines, primary, secondary, and tertiary monoamines with different substituents, as well as other interfering non-amines, were selected as guests (Fig. 8A). These guests with identical concentrations of 0.40 mM

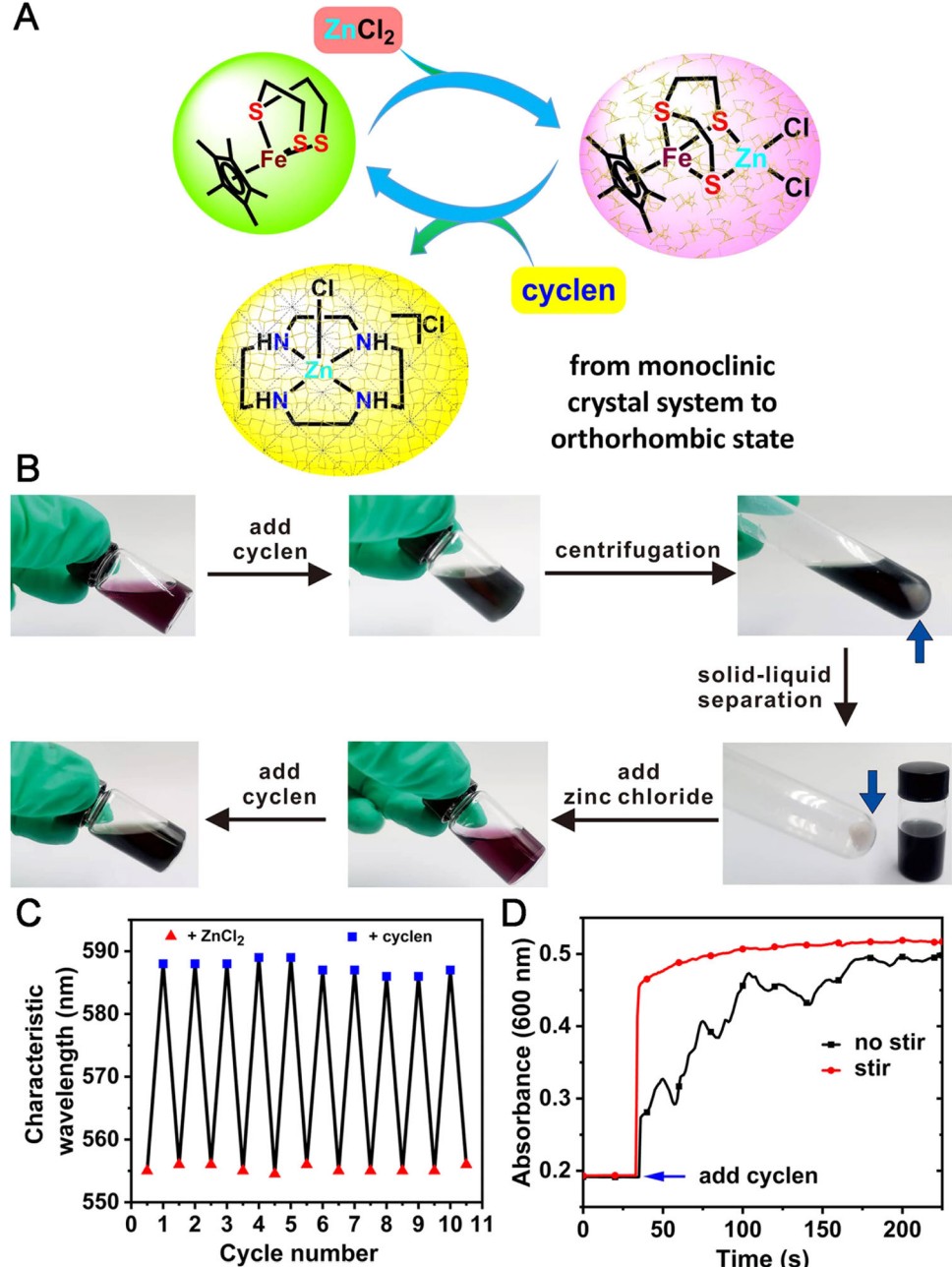

**Fig. 7 Cycle diagrams of C1 for detecting cyclen. A** Reaction mechanism of **C1** for the cyclic detection of cyclen. The ordered phase was maintained during the transition from **C1** to **N1**. **B** Operation illustration of **C1** (3.30 mM) for cyclic detection of cyclen (4.00 mM) in $CH_2Cl_2$ (3 mL). Obvious color changes between purplish-red and green were observed during the addition of cyclen or $ZnCl_2$. **C** Cycling experiments when **C1** (0.33 mM) was alternately treated by 0.40 mM cyclen or $ZnCl_2$ in $CH_2Cl_2$. The appearance of characteristic absorption peaks in the UV-vis spectra was used to judge the recovery of **C1** (585 nm) or [Cp*Fe($\eta^3$-tpdt)] (588 nm). **D** Time-dependence of absorbance intensity change (at 600 nm) upon the addition of cyclen (0.40 mM) to **C1** (0.33 mM) in $CH_2Cl_2$, with or without stirring at 20 rpm.

were added to $CH_2Cl_2$ solutions containing iron–sulfur complexes (0.33 mM), and then the massive UV-vis spectra were recorded (Supplementary Figs. S28–39). To analyze these spectral data and rank the responses of the complexes to organic amines, a statistical formula was developed (inset in Fig. 8B). The detection sensitivity was determined through the calculation of the integration area[58] of the UV-vis spectra between 380 and 750 nm before and after the interaction of a complex with a guest (Fig. 8B), and a larger integration area represents a greater response.

Based on the above calculation, the responsiveness of **C1**–**C3** toward various reagents are summarized in Fig. 8C–E. Complexes

**C1**–**C3** all showed no response to common volatile organic compounds (VOCs)[59,60], including aldehydes, ketones, esters, acids, alcohols, amides, and sulfides, and varying responses to different amine species. As shown in Fig. 8C, **C1** exhibited a strongest response to diethylenetriamine (**7**), followed by cyclen (**10**), a moderate response to chain polyamines (**1**-**6**, **8**, **9**), a weak response to other ployamines (**11**-**14**), and no response to all monoamines, which were consistent with the DPV data (black and red lines in Fig. 4H). Furthermore, **C2** exhibited a moderate to excellent response to all polyamines (except for 4,4'-bipyridine) and no response to monoamines (only one exception) (Fig. 8D).

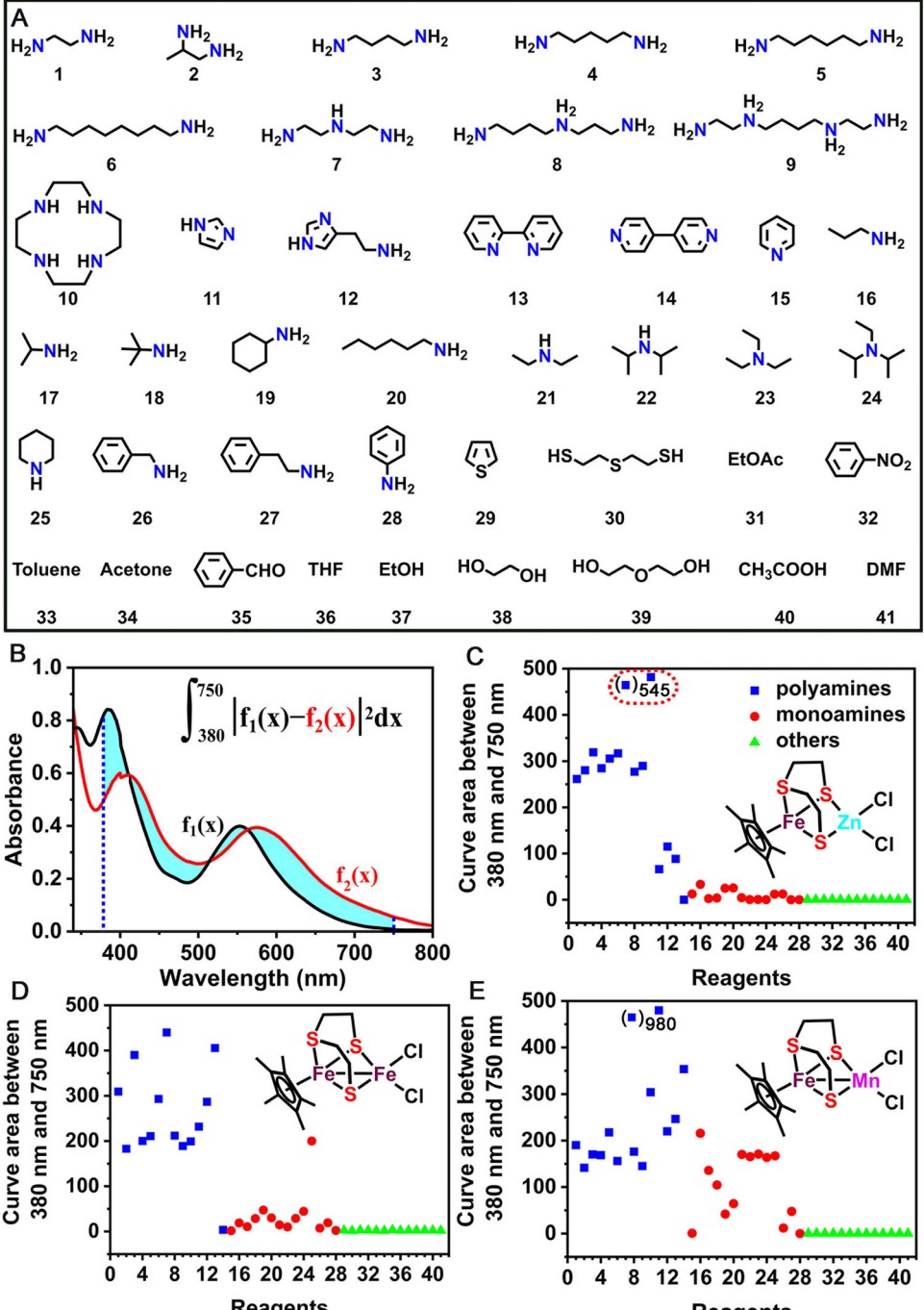

**Fig. 8 Comparison of the responses of complexes C1–C3 to different reagents according to UV-vis spectra. A** Chemical structures and labels of various polyamines, monoamines, and other interfering compounds. **B** The calculation method of the integration area of the UV-vis spectra between 380 and 750 nm, inset shows the calculation formula, a larger integration area represents a greater response to the guest. Summary of the positive and negative responses of complexes **C1** (**C**), **C2** (**D**), and **C3** (**E**) to the above reagents.

In contrast, **C3** showed a sensitive response to all polyamines and most of the monoamines listed in the table (Fig. 8E), which demonstrated its strong binding capacity toward amine species. Importantly, other polydentate ligands, such as tpdt, ethylene glycol, and diethylene glycol, were also tested and no responses were observed in Fig. 8C–E, which showed the specificity of our sensors for polyamines but not for polydentate ligands.

In brief, **C1** and **C2** enabled the selective detection of polyamines, while **C3** could detect amine species. Notably, **C1** could also achieve the detection of specific polyamines. The

specificity of the above array for polyamines as well as amine species showed the superiority of our developed sensors based on iron–sulfur moiety. Although **C1**, **C2**, and **C3** possessed analogous binuclear structures, there are large differences in the ability for polyamines detection, which may be elaborated from the following two points: **1**, the different species of the second metal ions ($Zn^{2+}$, $Fe^{2+}$, $Mn^{2+}$) led to differences in their coordination abilities with various reagents, but they all had a strong interaction with amine species relative to other reagents; **2**, compared to single dentate ligands, multidentate ligands were

**Table 1 Comparison between our work with other polyamines detection probes.**

| probe | response time | duplication | mechanism | refs |
|---|---|---|---|---|
| iron–sulfur complexes | second level | recyclability | order–order assembly transition | this work |
| PPAB-TPE | 3 min | disposability | AIE effect | 10 |
| CVDA dye | 30 min | disposability | assembly disorder order transition | 18 |
| Cu(II) complex | few hours | disposability | coordination binding | 21 |
| FCP-Pd | 30 min | disposability | coordination binding | 57 |
| CDs/CdTe QDs | 1 min | disposability | N/A | 61 |
| DSB derivatives | slowly | disposability | chemical reaction | 62 |
| TPE-pentiptycene | 5 min | disposability | electrostatic pairing and hydrogen bonding | 63 |

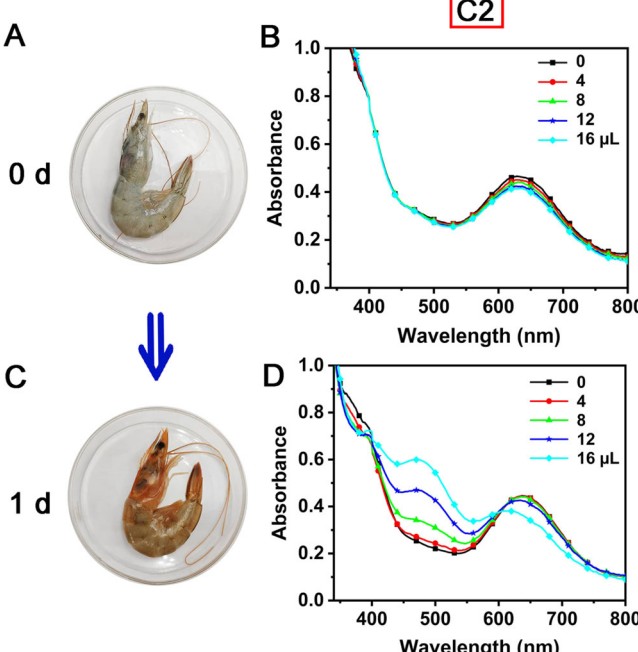

**Fig. 9 Monitoring fresh shrimp at 27 °C using our developed approach.** Photos of fresh shrimp (**A**) and shrimp left for one day (**C**). **B**, **D** UV-vis spectra of **C2** after interaction with exudates of shrimp.

more accessible to metal ions to form more stable metal complexes and thus had more opportunities for competition. Moreover, the introduction of the second metal ions endowed these iron–sulfur complexes with unparalleled properties. The comparison between our work with other polyamine detection probes[61–63], including molecular probes, chemosensors, and nanosensors was summarized in Table 1, which further confirmed the superiority of our work in polyamines detection.

Furthermore, as for the trinuclear complexes **C4**–**C6**, a credible response could not be detected for all of the analytes (Supplementary Fig. S40), which was attributed to the fact that their large spacial hindrance and saturated coordination environment of the three metal centers [FeMFe] (M = Pd, Ni, Cu) led to the inaccessibility of the above analytes.

Additionally, to affirm the practical utility of our developed approach, shrimp spoilage was successfully monitored by **C1**–**C3** using UV-vis spectra (Fig. 9, Supplementary Fig. S41). Initially, none of the UV-vis spectra of **C1**–**C3** changed significantly in the presence of fresh shrimp exudate (Fig. 9B, Supplementary Fig. S41B, C). After leaving the fresh shrimp at 27 °C overnight, the shrimp turned red overall (Fig. 9C) and produced an exudate with a pungent odor. Surprisingly, **C1**–**C3** all showed varying degrees of responses to different volumes of these exudates (Fig. 9D, Supplementary Fig. S41E, F) and the spectral trend was

roughly consistent with their interaction with pure polyamines. The above results indicated that our developed method could analyze complex samples containing polyamines and also demonstrated the potential application of our method in monitoring food freshness.

**The universality of the sensing mechanism.** The order–order assembly transition-driven polyamines detection phenomenon was not only observed in the reaction between **C1** and cyclen (Supplementary Fig. S54B, C), other polyamines could also induce the decomposition of **C1** into mononuclear [Cp*Fe($\eta^3$-tpdt)] and the formation of original assemblies. For example, $Zn^{2+}$−ethylenediamine self-assembled into a petaloid structure with an average size of 42 μm (Supplementary Figs. S54D, S55A), while $Zn^{2+}$−4,4'-bipyridine formed numerous urchin-like structures with an average size of 5.6 μm (Supplementary Figs. S54E, S55B).

Furthermore, binuclear iron-iron complex **C2** and iron-manganese complex **C3** exhibited a comparable behaved behavior to **C1**, also displaying high selectivity toward polyamines superior to monoamines and other interfering non-amines. The spatial packing mode of **C2** was also analogous to that of **C1**, the molecules were connected by intermolecular H···Cl bonds (Supplementary Figs. S4, S56A). When **C2** interacted with 2,2'-bipyridine, **C2** decomposed to mononuclear [Cp*Fe($\eta^3$-tpdt)] and generated [($\eta^2$-2,2'-bipyridine)FeCl$_2$] (**N2**). The detailed crystal information of **N2** (CCDC 2178630, see Supplementary Data 4, 9, Figs. S12, S13, and Table S10 for details), NMR spectra (Supplementary Figs. S19, S20), and FT-IR data (Supplementary Fig. S52) of **N2** are smoothly obtained. **N2** could self-assembled into numerous hexagon nanosheets (Supplementary Figs. S54F, S55C, and S55D), and its molecular structure and spatial accumulation were shown in Supplementary Fig. S55E–G, respectively. Other patterns (Supplementary Fig. S55H–J) further validated the diversity of self-assemblies of $Fe^{2+}$−polyamine complexes. Meanwhile, the repeatability of **C2** for cyclen detection was guaranteed (Supplementary Fig. S59A).

A analogous effect could be observed in binuclear iron-manganese complex **C3** (Supplementary Fig. S56B). Rectangular laminas (Supplementary Fig. S56C) with an average length of 14.4 μm and width of 6.2 μm were formed in the presence of diethylenetriamine (Supplementary Fig. S56D, E), while wafer-like self-assemblies with varying diameters of $Mn^{2+}$−iminazole complex are depicted in Supplementary Fig. S56F. $Mn^{2+}$−cyclen complex [($\eta^4$-cyclen)MnCl][Cl] (**N3**, CCDC 2178629, see Supplementary Data 5 and 10 for details) could self-assemble into cubes (Supplementary Fig. S56G) with an average length of 6.2 μm (Supplementary Fig. S56H) and a high crystallinity was determined *via* X-ray method (Supplementary Fig. S56I). The molecular structure (Supplementary Fig. S14) and spatial packing (Supplementary Fig. S15) of **N3** were analogous to those of **N1**, except for a slight difference in M−N bond and hydrogen bond length. The main bond lengths and bond angles, EPR data, ESI-

HRMS data, and FT-IR data of **N3** are shown in Supplementary Table S11, Figs. S26, S46, and S53, respectively. Furthermore, the repeatability of **N3** for cyclen detection was also guaranteed (Supplementary Fig. S59B).

## Discussion

In this paper, a variety of environmentally stable iron–sulfur sensors had been developed. The integration of iron–sulfur skeleton and different transition metals provided a versatile sensing platform for the specific, sensitive, and rapid detection of polyamines, while clearly discriminating polyamines from monoamines and other interfering compounds. This platform allowed for immediate polyamines discrimination by distinct color changes visible to the naked eyes, and a low detection limit was achieved by the DPV method. Hence, more portable, automatic, and precise devices suitable for selective polyamines detection could be fabricated by integrating chemically inert and transparent substrates (such as poly(dimethylsiloxane))[64] and measured by the euclidean distance of RGB values[65].

Importantly, an innovative polyamines response mode driven by an order–order transition of the assemblies was smoothly revealed, which facilitated a range of superior detection effects, including specific and selective polyamine detection, second-level response, low detection limit, visible colorimetric detection, and good recyclability of these iron–sulfur sensing materials. Moreover, behind the order–order transition of the assemblies was the generation of massive metal–polyamines complexes. Given the great applications of metal–polyamines complexes in multifarious fields ranging from material chemistry to chemical biology[66,67], several synthetic methods had been explored[68–71], but our work provided a simpler, faster, and more accurate synthetic method to build this class of products. Amazingly, better crystallinity (Supplementary Fig. S57) and high-specific assemblies were achieved (Supplementary Fig. S58). In brief, this work confirmed the application of iron–sulfur complexes in the analytical field, and provide a paradigm for their development in a wider range of applications[72,73], such as the detection of sugars, pesticides, fungi, thiols, etc.

## Methods

**Preparation of [Cp\*Fe($\mu$-1$_\kappa^3$SSS':2$_\kappa^2$SS-tpdt)ZnCl$_2$] (C1)**. In glove-box, solid ZnCl$_2$ (42 mg, 0.31 mmol) was added to mononuclear [Cp\*Fe($\eta^3$-tpdt)] precursor (103 mg, 0.30 mmol) in CH$_2$Cl$_2$ (10 mL) at room temperature, the resulting solution was stirred for about 0.5 h. During this time, the solution changed gradually from green to purplish red. Then the mixture was filtered, evaporated, and washed with THF (3 × 5 mL). The product, [Cp\*Fe($\mu$-1$_\kappa^3$SSS':2$_\kappa^2$SS-tpdt)ZnCl$_2$] (C1, 137 mg, 0.29 mmol, 95%), was obtained as a deep purple crystalline powder. Crystals suitable for X-ray diffraction were obtained from a CH$_2$Cl$_2$ solution layered with nhexane at room temperature. $^1$H NMR (400 MHz, CD$_2$Cl$_2$, ppm): $\delta$ −9.91 (br, 8H, tpdt-C$H_2$), −32.24 (br, 15H, Cp\*-C$H_3$). EPR (298 K): g = 2.10. IR (film, cm$^{-1}$): 2972, 2919, 1478, 1376, 1276, 1157, 1023, 929, 839.

**Preparation of [Cp\*Fe($\mu$-1$_\kappa^3$SSS':2$_\kappa^2$SS-tpdt)FeCl$_2$] (C2)**. The synthetic path of **C2** was similar to that of **C1**, except for changing ZnCl$_2$ to FeCl$_2$(THF)$_{1.5}$. Complex [Cp\*Fe($\mu$-1$_\kappa^3$SSS':2$_\kappa^2$SS-tpdt)FeCl$_2$] (C2, 90%) was obtained as a deep green crystalline powder. $^1$H NMR (400 MHz, CD$_2$Cl$_2$, ppm): $\delta$−10.22 (br, 8H, tpdt-C$H_2$), −21.49 (br, 15H, Cp\*-C$H_3$). EPR (298 K): g = 2.02.

**Preparation of [Cp\*Fe($\mu$-1$_\kappa^3$SSS':2$_\kappa^2$SS-tpdt)MnCl$_2$] (C3)**. The synthetic path of **C3** was similar to that of **C1**, except for changing ZnCl$_2$ to MnCl$_2$. Complex [Cp\*Fe($\mu$-1$_\kappa^3$SSS':2$_\kappa^2$SS-tpdt)MnCl$_2$] (C3, 92%) was obtained as a deep purple crystalline powder. Crystals suitable for X-ray diffraction were obtained from a CH$_2$Cl$_2$ solution layered with $^n$hexane at room temperature. $^1$H NMR (400 MHz, CD$_2$Cl$_2$, ppm): $\delta$ 39.38 (br, 4H, tpdt-C$H_2$), 19.78 (br, 4H, tpdt-C$H_2$), −21.31 (br, 7.5H, Cp\*-C$H_3$), −36.80 (br, 7.5H, Cp\*-C$H_3$). EPR (298 K): g = 2.08. IR (film, cm$^{-1}$): 2956, 2920, 2854, 1724, 1476, 1426, 1375, 1275, 1072, 1022, 928, 828.

**Preparation of [Cp\*Fe($\mu$-1$_\kappa^3$SSS':2$_\kappa^2$SS-tpdt)Pd($\mu$-2$_\kappa^2$SS:3$_\kappa^3$SSS'-tpdt) FeCp\*][PF$_6$]$_2$ (C4)**. In glove-box, solid PdCl$_2$ (27 mg, 0.15 mmol) was added to complex [Cp\*Fe($\eta^3$-tpdt)] precursor (103 mg, 0.30 mmol) in CH$_3$OH (10 mL) at room temperature and stirred for about 2 h. The solution changed gradually from green to brown-red. Then NH$_4$PF$_6$ (197 mg, 1.20 mmol) was added to the resultant solution as a solid whereupon immediate formation of a purple precipitate was observed. The residue was collected, extracted with CH$_3$CN, evaporated, and washed with Et$_2$O (3 × 5 mL). The product, [Cp\*Fe($\mu$-1$_\kappa^3$SSS':2$_\kappa^2$SS-tpdt)Pd($\mu$-2$_\kappa^2$SS:3$_\kappa^3$SSS'-tpdt)FeCp\*][PF$_6$]$_2$ (C4, 151 mg, 0.14 mmol, 90%) was obtained as a deep yellow crystalline powder. $^1$H NMR (400 MHz, CD$_3$CN, ppm): $\delta$−2.51 (br, 4H, tpdt-C$H_2$), −3.83 (br, 4H, tpdt-C$H_2$), −17.76 (br, 4H, tpdt-C$H_2$), −21.95 (br, 4H, tpdt-C$H_2$), −23.68 (br, 15H, Cp\*-C$H_3$), −29.62 (br, 15H, Cp\*-C$H_3$). ESI-HRMS (m/z): calcd for [**C4**−2PF$_6$]$^{2+}$: 395.9835; found: 395.9830. EPR (298 K): g = 2.06.

**Preparation of [Cp\*Fe($\mu$-1$_\kappa^3$SSS':2$_\kappa^2$SS-tpdt)Ni($\mu$-2$_\kappa^2$SS:3$_\kappa^3$SSS'-tpdt)FeCp\*] [PF$_6$]$_2$ (C5)**. The synthetic path of **C5** was similar to that of **C4**, except for changing PdCl$_2$ to NiCl$_2$. Complex [Cp\*Fe($\mu$-1$_\kappa^3$SSS':2$_\kappa^2$SS-tpdt)Ni($\mu$-2$_\kappa^2$SS:3$_\kappa^3$SSS'-tpdt)-FeCp\*][PF$_6$]$_2$ (C5, 91%) was obtained as a deep brown crystalline powder. $^1$H NMR (400 MHz, CD$_2$Cl$_2$, ppm): $\delta$ 3.31 (br, 8H, tpdt-C$H_2$), 2.93 (br, 8H, tpdt-C$H_2$), 1.58 (s, 30H, Cp\*-C$H_3$). ESI-HRMS (m/z): calcd for [**C5**−2PF$_6$]$^{2+}$: 371.9989; found: 371.9991.

**Preparation of [Cp\*Fe($\mu$-1$_\kappa^3$SSS':2$_\kappa^2$SS-tpdt)Cu($\mu$-2$_\kappa^2$SS:3$_\kappa^3$SSS'-tpdt) FeCp\*][PF$_6$] (C6)**. The synthetic path of **C6** was similar to that of **C4**, except for changing PdCl$_2$ to CuCl. Complex [Cp\*Fe($\mu$-1$_\kappa^3$SSS':2$_\kappa^2$SS-tpdt)Cu($\mu$-2$_\kappa^2$SS:3$_\kappa^3$SSS'-tpdt)FeCp\*][PF$_6$] (C6, 92%) was obtained as a deep purple crystalline powder. $^1$H NMR (400 MHz, CD$_2$Cl$_2$, ppm): $\delta$ 7.31 (s, 4H, tpdt-C$H_2$), −10.94 (m, 4H, tpdt-C$H_2$), −14.14 (s, 4H, tpdt-C$H_2$), −20.05 (s, 4H, tpdt-C$H_2$), −29.96 (s, 30H, Cp\*-C$H_3$). ESI-HRMS (m/z): calcd for [**C6**−PF$_6$]$^+$: 748.9921; found: 748.9923. EPR (298 K): g = 2.06.

**Preparation of [($\eta^4$-cyclen)ZnCl][Cl] (N1)**. At room temperature, a solution of cyclen (43 mg, 0.25 mmol) was added to a CH$_2$Cl$_2$ solution of **C1** (101 mg, 0.21 mmol) and stirred vigorously for about 10 min, resulting in a green solution. Then the supernatant solution was centrifuged and the residue was washed with CH$_2$Cl$_2$ (3 × 3 mL). After evaporation to drying in a vacuum, the product [($\eta^4$-cyclen)ZnCl][Cl] (N1, 61 mg, 0.20 mmol, 95%) was obtained as a light yellow solid. Crystals suitable for X-ray diffraction were obtained from a CH$_3$OH solution layered with Et$_2$O at room temperature. $^1$H NMR (400 MHz, D$_2$O, ppm): $\delta$ 3.05 (s, 4H, N$H$), 2.89-2.94 (m, 8H, cyclen-C$H_2$), 2.75–2.81 (m, 8H, cyclen-C$H_2$). $^{13}$C NMR (100 MHz, D$_2$O): $\delta$ 43.60 (cyclen-C$H_2$). ESI-HRMS (m/z): calcd for [**N1**−Cl]$^+$: 271.0668; found: 271.0664. IR (film, cm$^{-1}$): 3243, 3167, 2919, 2870, 1486, 1091, 1011, 807.

**Preparation of [($\eta^2$-2,2′-bipyridine)FeCl$_2$] (N2)**. At room temperature, a solution of 2,2'-bipyridine (39 mg, 0.25 mmol) was added to a CH$_2$Cl$_2$ solution of **C2** (100 mg, 0.21 mmol) and stirred vigorously for about 10 min, resulting in a red solution. Then the supernatant solution was evaporated, washed with Et$_2$O (3 × 5 mL), and dried in a vacuum. The product [($\eta^2$-2,2'-bipyridine)FeCl$_2$] (N2, 55 mg, 0.20 mmol, 93%) was obtained as a dark red solid. Crystals suitable for X-ray diffraction were obtained from a CH$_3$OH solution layered with Et$_2$O at room temperature. $^1$H NMR (400 MHz, DMSO-$d_6$, ppm): $\delta$ 8.84 (s, 2H, Ar-H), 8.17 (s, 2H, Ar-H), 7.36–7.49 (m, 4H, Ar-H). $^{13}$C NMR (100 MHz, DMSO-$d_6$): $\delta$ 157.64, 152.80, 137.94, 126.84, 123.40. IR (film, cm$^{-1}$): 3059, 1601, 1464, 1441, 1427, 773, 734.

**Preparation of [($\eta^4$-cyclen)MnCl][Cl] (N3)**. At room temperature, a solution of cyclen (43 mg, 0.25 mmol) was added to a CH$_2$Cl$_2$ solution of **C3** (100 mg, 0.21 mmol) and stirred vigorously for about 10 min, resulting in a green solution. Then the supernatant solution was evaporated, washed with CH$_2$Cl$_2$ (3 × 3 mL), and dried in a vacuum. The product [($\eta^4$-cyclen)MnCl][Cl] (N3, 56 mg, 0.19 mmol, 90%) was obtained as a yellow solid. Crystals suitable for X-ray diffraction were obtained from a CH$_3$OH solution layered with Et$_2$O at room temperature. ESI-HRMS (m/z): calcd for [**N3**−Cl]$^+$: 262.0757; found: 262.0739. EPR (298 K): g = 2.00. IR (film, cm$^{-1}$): 3508, 3459, 3242, 2904, 2861, 1645, 1443, 1359, 1096, 1006, 928, 847.

## Data availability

Detailed experimental details are available in the Supplementary Methods. The single-crystal data generated in this study have been deposited in The Cambridge Crystallographic Data Center under accession code CCDC-2171251 (for **C1**, Supplementary Data 1), CCDC-2171252 (for **C3**, Supplementary Data 2), CCDC-2171267 (for **N1**, Supplementary Data 3), CCDC-2178630 (for **N2**, Supplementary Data 4), and CCDC-2178629 (for **N3**, Supplementary Data 5). Their checkcif files can be found in Supplementary Data 6, Data 7, Data 8, Data 9, and Data 10, respectively. These data can be obtained free of charge from The Cambridge Crystallographic Data Center via www.ccdc.cam.ac.uk/data_request/cif. $^1$H and $^{13}$C NMR spectra can be found in Supplementary NMR Spectra. The video of the recyclability of **C1** in cyclen detection assays can be found in Supplementary Movie S1.

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

## Acknowledgements
This work was supported by the National Key R&D Program of China (Grant No. 2022YFC3400800), the National Natural Science Foundation of China (21922411 and 22174138), DICP Innovation Funding (DICP-RC201801, I202008, I202243, and I202229), the Dalian Outstanding Young Scientific Talent (2020RJ01), and Liaoning Province Doctoral Initiation Fund (2021-BS-009).

## Author contributions
Y.Z. conceived the concept, performed experiments, and prepared the paper and Supplementary Information. X.Z. conceived the concept. Y.Q. explored the detection substrate scope. X.L., Y.C., Z.S., M.S., W.S., J.X., and Z.L. discussed the results. G.Q. conceived and directed the project, contributed to the writing of the paper, discussed over the results and commented on the paper.

## Competing interests
The authors declare no competing interests.
