## [Peer Review File · Communications Chemistry]

Reviewers' comments:

Reviewer #1 (Remarks to the Author):

The manuscript describes new principle of polyamines detection using their ability to form very stable complexes with metal ions that induces structural transition in colored Fe-S complexes. The sensor response mechanism and related structural changes are deeply investigated by numerous methods, described in details and undoubtedly have novelty and significance. Although the effects are interesting, the sensing task itself is not very clearly stated and proved. In my opinion it should be explained what are types of samples to be analyzed using this approach, and what are the minimal allowed concentrations of the polyamines in comparison with LOD reached? Will it be possible to use this approach to analyze the mixture of polyamines and polyamines in the presence, for example, humic substances (HS), in surface waters analysis.

Reviewer #2 (Remarks to the Author):

Zhang et al reported "Assembly Order-Order Transition-Driven Polyamines Detection 2 Based on Iron-Sulfur Complexes with unique mechanism and good recyclability". The topic is important and interesting. The manuscript is well written. However, there are some suggestions that would like to be considered before then.

1 For polyamine detection, the typical polyamines, monoamine and diamines such as spermine, spermidine, ethylenediamine, propylenediamine, putrescine, cadaverine, hexamethylenediamine, octamethylenediamine, n-propyl amine, n-hexylamine, cyclohexylamine, aniline should be selected in selectivity and anti-interference test.

2 The advantages of polyamine detection in this work should be compared with other reported references and summarized in a Table.

3 Are these metal complexes and N1 emissive? Do they have any fluorescence?

4 What's the structure-detection performance relationship after comparison of the responses of complexes C1-C3 to different reagent? The detailed discussion should be added.

5 Chromophore Reaction-based Fluorescent Probes for polyamines Detection (Journal of Materials Chemistry C, 2022, 10, 5672-5683; Biosensors, 2022, 12, 633; Journal of Materials Chemistry B, 2021, 9, 9383-9394; Dyes and Pigments, 2021, 194, 109634, Sensors and Actuators B: Chemical, 2020, 312, 127953, Analytica Chimica Acta 1135 (2020) 38-46) should be cited in Introduction section.

Reviewer #3 (Remarks to the Author):

The author introduced a new sensing mode for polyamine detection based on assembly order-order transition-driven iron-sulfur complexes. This unique mechanism enhances detection efficiency, specificity, and recyclability, offering rapid, visual detection. This study also provides a paradigm for

applying iron-sulfur platforms in environment-related fields. Overall, this study is novel and interesting to the community and I recommend the paper to publish before addressing these minor revisions:

1. For the synthesis and characterization section, the paragraph could benefit from some reformatting to improve readability. Breaking it into shorter paragraphs and grouping related information together could help. For instance, consider separating information about the synthesis process from information about the characterization of the complexes. Also, when discussing the UV-vis spectra, it would be helpful to mention the specific wavelengths or ranges of the absorption bands for a clearer understanding of the results.

2. In some instances, you used abbreviations without introducing them first. Make sure to introduce any abbreviations before using them. For example, "EtOAc" should be introduced as "ethyl acetate (EtOAc)" before using the abbreviation.

3. When describing the "Tyndall" effect, it might be helpful to briefly explain what it is for readers who may not be familiar with the term. For example: "which were confirmed by the "Tyndall" effect, a light scattering phenomenon observed in colloidal suspensions.

4. It would be good if the authors could elaborate on the significance of the differences in structural composition and spatial packing mode between C1 and N1. Discussing the implications of these differences may provide more context and enhance the reader's comprehension.

5. Some information appears repetitive or redundant, particularly in discussing the detection responses of various complexes (C1, C2, C3) towards different amines. Consider summarizing these responses in a more concise manner or presenting them in a tabular format to improve clarity.

Response to referees:

We thank the referees for their thoughtful comments, and hope they are doing well during this pandemic situation. In response to the referees' suggestions, we have added the related experiments and made the corresponding modifications to the manuscript. The referee's comments are italicized, and our responses immediately follow.

Point-by-point responses to the comments and concerns of Referee # 1

Comment 1: In my opinion it should be explained what are types of samples to be analyzed using this approach.

Response: Thanks for your advice! To further summarize the sample types to be analyzed by our developed method, a total of 41 compounds in three categories, including chain and ring polyamines, primary, secondary, and tertiary monoamines with different substituents, as well as other interfering non-amines, were selected as guests (Fig. 8A in the revised manuscript). Through the collection and comparison of extensive UV-vis data (Fig. S28–39 in the revised SI) after treating complexes **C1–C6** with various reagents, we found that both **C1** and **C2** can achieve the selective detection of polyamines, while **C3** enables the detection of both polyamines or monoamines, and **C4–C6** show no response to all reagents. Furthermore, freshness monitoring tests of shrimp demonstrate that **C1–C3** can analyze complex samples containing polyamines (Fig. S41 in the revised SI). A detailed and summarized description was provided in the “The Specificity of Polyamines Detection” section.

Comment 2: what are the minimal allowed concentrations of the polyamines in comparison with LOD reached?

Response: Thanks for your question! As shown in Fig. S47A in the revised SI, DPV data for **C1** in the presence of different concentrations (0.3, 3, 30, 60, 150, 300, 600, 900, 1200, 1500, and 1800 μ M) of cyclen are collected. Compared to the silence in the presence of 0.3 μ M cyclen, the current value of **C1** at -0.96 V begins to decrease slightly when 3 μ M cyclen is added, which is consistent with the detection limit of the micromolar level reached by LOD. Relevant text descriptions are added in the “Amperometric Detection of Polyamines” section and marked in red.

Comment 3: Will it be possible to use this approach to analyze the mixture of polyamines and polyamines in the presence, for example, humic substances (HS), in surface waters analysis.

Response: Thanks for your suggestions! Taking shrimp as an example (Fig. 9 and Fig. S41), we collect the UV-vis spectra of C1–C3 after the interaction with the exudates of fresh or spoiled shrimp and found that complexes C1–C3 all showed varying degrees of responses to different volumes of exudates. These results were roughly consistent with their interaction with pure polyamines and also indicated that our developed method could analyze complex samples containing polyamines. Detailed text descriptions are added in the “The Specificity of Polyamines Detection” section.

Fig. 9 and Fig. S41. Monitoring fresh shrimp at 27 °C using our developed approach. (A, E) Photos of fresh shrimp (A) and shrimp left for one day (E). (B–D, F–H) UV-vis spectra of C1 (B, F), C2 (C, G) and C3 (D, H) after interaction with exudates of shrimp.

Point-by-point responses to the comments and concerns of Referee # 2

Comment 1: For polyamine detection, the typical polyamines, monoamine and diamines such as spermine, spermidine, ethylenediamine, propylenediamine, putrescine, cadaverine, hexamethylenediamine, octamethylenediamine, n-propyl amine, n-hexylamine, cyclohexylamine, aniline should be selected in selectivity and anti-interference test.

Response: Thanks for your suggestions! In the revised manuscript, a total of 41 compounds (Fig. 8A in the revised manuscript) are selected as guests, including the above objects mentioned by the reviewer. These results indicate that **C1** and **C2** enable the detection of polyamines, while **C3** can detect amine species (Fig. 8C–E in the revised manuscript). Notably, **C1** can also achieve selective detection of specific polyamines. As for the trinuclear complexes **C4–C6** (Fig. S40 in the revised SI), no credible responses can be detected in the presence of all analytes. Detailed test results and text description are added in the “The Specificity of Polyamines Detection” section.

Comment 2: The advantages of polyamine detection in this work should be compared with other reported references and summarized in a Table.

Response: Thanks for your advice! In the previous work, we fully present the superiority of our method for polyamines detection, and indeed ignore the comparison with the reported probes. In the revised manuscript, the comparison between our work and other polyamine detection probes, including molecular probes, chemosensors, and nanosensors were summarized in Table 1.

Table 1 | Comparison between our work with other polyamines detection probes

probe	response time	duplication	mechanism	refs
iron–sulfur complexes	second level	recyclability	assembly order-order transition	this work
PPAB-TPE	3 minutes	disposability	AIE effect	10
CVDA dye	30 minutes	disposability	assembly disorder order transition	18
Cu(II) complex	few hours	disposability	coordination binding	21
FCP-Pd	30 minutes	disposability	coordination binding	57
CDs/CdTe QDs	1 minute	disposability	N/A	61
DSB derivatives	slowly	disposability	chemical reaction	62
TPE-pentiptycene	5 minutes	disposability	electrostatic pairing and hydrogen bonding	63

Comment 3: Are these metal complexes and N1 emissive? Do they have any fluorescence?

Response: Thanks for your question! Using the strong UV-vis absorption peaks of complexes C1–C6 and N1 as excitation wavelengths, their fluorescence spectra are collected smoothly and no obvious fluorescence absorption peaks are observed. The detailed test results are shown below.

(A–I) Emission spectra of C1 (A, B), C2 (C, D), C3 (E, F), C4 (G), C5 (H), and C6 (I) at different excitation wavelengths in CH₂Cl₂. The concentrations of C1–C6 were all at 0.33 mM. (J–L) Absorption (J) and emission (K, L) spectra of N1 (0.33 mM) in CH₃OH.

Comment 4: What's the structure-detection performance relationship after comparison of the responses of complexes C1–C3 to different reagent? The detailed discussion should be added.

Response: Thanks for your suggestion! After the elaboration on the responses of **C1**, **C2**, and **C3** to various reagents, we have summarized their differences in organic amines detection and analyzed the possible reasons for the selective polyamines detection in terms of structure-detection performance relationship. The detailed text description is presented in the “The Specificity of Polyamines Detection” section of the revised manuscript and marked in red.

Comment 5: Chromophore Reaction-based Fluorescent Probes for polyamines Detection (Journal of Materials Chemistry C, 2022, 10, 5672-5683; Biosensors, 2022, 12, 633; Journal of Materials Chemistry B, 2021, 9, 9383-9394; Dyes and Pigments, 2021, 194, 109634, Sensors and Actuators B: Chemical, 2020, 312, 127953, Analytica Chimica Acta 1135 (2020) 38-46) should be cited in Introduction section.

Response: We thank the reviewer for pointing out our omission of the papers on polyamines detection with chromophore reaction-based fluorescent probes. All papers have been cited in the Introduction section and marked in red.

Point-by-point responses to the comments and concerns of Referee # 3

Comment 1: For the synthesis and characterization section, the paragraph could benefit from some reformatting to improve readability. Breaking it into shorter paragraphs and grouping related information together could help. For instance, consider separating information about the synthesis process from information about the characterization of the complexes. Also, when discussing the UV-vis spectra, it would be helpful to mention the specific wavelengths or ranges of the absorption bands for a clearer understanding of the results.

Response: Thanks for your suggestion and sorry for the inconvenience to you caused by our descriptions. For the synthesis and characterization section, we have integrated the relevant information about these iron–sulfur complexes and separated the synthesis method from their characterization data. Additionally, the long paragraph has been broken up and some descriptions are also refined.

For the discussion section of UV-vis spectra, the comparison of the precursor [Cp*Fe(η^3 -tpdt)] before and after the introduction of the second metal ions are presented in detail, including the changes of the absorption peaks and the specific shift of the wavelength. All the changes are marked in red.

Comment 2: In some instances, you used abbreviations without introducing them first. Make sure to introduce any abbreviations before using them. For example, "EtOAc" should be introduced as "ethyl acetate (EtOAc)" before using the abbreviation.

Response: Thanks for your comments and sorry for the inconvenience to you caused by our negligence. In the revised manuscript, the abbreviations have been introduced in their first citation and marked in red, including tpdt, M, and EtOAc.

Comment 3: When describing the "Tyndall" effect, it might be helpful to briefly explain what it is for readers who may not be familiar with the term. For example: "which were confirmed by the "Tyndall" effect, a light scattering phenomenon observed in colloidal suspensions.

Response: We are grateful for the suggestion that allows us to make our article more readable. According to the review's advice, we have added an acceptable description of the "Tyndall" effect and marked it in red.

Comment 4: It would be good if the authors could elaborate on the significance of the differences in structural composition and spatial packing mode between C1 and N1. Discussing the implications of these differences may provide more context and enhance the reader's comprehension.

Response: Thanks for your suggestion! In the "Assembly Order-Order Transition" section, we have performed a detailed analysis of **C1** and **N1** separately in terms of structural composition and spatial packing mode, but there are no further differential comparisons during this transformation process. We apologize for any distress caused to the reader. In the revised manuscript, at the end of this section, the differences in structural and spatial packing information between **C1** and **N1**, as well as the structural changes during the transition from **C1** and **N1** are described and marked in red.

Comment 5: Some information appears repetitive or redundant, particularly in discussing the detection responses of various complexes (C1, C2, C3) towards different amines. Consider summarizing these responses in a more concise manner or presenting them in a tabular format to improve clarity.

Response: We very strongly agree with your suggestion! In the "Responsive Behaviors to Polyamines" section, we have refined the written description of the response results of various complexes (**C1**, **C2**, and **C3**) towards different amines and removed some redundant statements. Similarly, we also condensed the other repetitive information in the revised manuscript. All changes are marked in red.

REVIEWERS' COMMENTS:

Reviewer #1 (Remarks to the Author):

I think that revised manuscript can be published. It was substantially improved and the authors have provided relevant comments to most of the points addressed in review process.

Reviewer #2 (Remarks to the Author):

The authors have answered all questions.

Reviewer #3 (Remarks to the Author):

The revised paper has resolved all my comments and recommendations so I recommend it be published.